# Stars: Tera-Scale Graph Building for Clustering and Graph Learning

**CJ Carey**
Google Research
cjcarey@google.com

**Jonathan Halcrow**
Google Research
halcrow@google.com

**Rajesh Jayaram**
Google Research
rkjayaram@google.com

**Vahab Mirrokni**
Google Research
mirrokni@google.com

**Warren Schudy**
Google Research
wschudy@google.com

**Peilin Zhong**
Google Research
peilinz@google.com

## Abstract

A fundamental procedure in the analysis of massive datasets is the construction of similarity graphs. Such graphs play a key role for many downstream tasks, including clustering, classification, graph learning, and nearest neighbor search. For these tasks, it is critical to build graphs which are sparse yet still representative of the underlying data. The benefits of sparsity are twofold: firstly, constructing dense graphs is infeasible in practice for large datasets, and secondly, the runtime of downstream tasks is directly influenced by the sparsity of the similarity graph. In this work, we present *Stars*: a highly scalable method for building extremely sparse graphs via two-hop spanners, which are graphs where similar points are connected by a path of length at most two. Stars can construct two-hop spanners with significantly fewer similarity comparisons, which are a major bottleneck for learning based models where comparisons are expensive to evaluate. Theoretically, we demonstrate that Stars builds a graph in nearly-linear time, where approximate nearest neighbors are contained within two-hop neighborhoods. In practice, we have deployed Stars for multiple data sets allowing for graph building at the *Tera-Scale*, i.e., for graphs with tens of trillions of edges. We evaluate the performance of Stars for clustering and graph learning, and demonstrate 10~1000-fold improvements in pairwise similarity comparisons compared to different baselines, and 2~10-fold improvement in running time without quality loss.

## 1  Introduction

Given a collection of unlabeled data, a critical technique in many data mining and machine learning pipelines is the construction of a similarity graph over the data. Similarity graphs provide a sparse representation of the underlying structure of a dataset by creating edges between the most similar data points. This allows downstream analytic procedures to run more efficiently over the data by restricting comparisons to edges in the graph. In particular, many such procedures, such as Hierarchical Agglomerative Clustering (HAC) and variants [17, 5, 1], can be run in time nearly-linear in the number of edges.

In order to be useful as part of a larger pipeline, graph building itself must be an extremely efficient procedure. Namely, if the time required to construct the graph exceeds the size of the graph itself, then graph building may become the primary bottleneck of the pipeline. To avoid this scenario, graph building algorithms must circumvent all-pairs comparisons by employing selective filtering methods, such as *locality sensitive hashing* (LSH). However, with the extraordinary scale of modern datasets, LSH alone is often no longer sufficient to ensure efficiency. Such methods still require an all-pairs

comparison within each hash bucket, which is infeasible for the bucket sizes produced by LSH on large datasets. This is not deficit of LSH specifically, but is rather an intrinsic dilemma of scale: even ground truth similarity graphs, such as threshold graphs and $k$-nearest neighbor graphs, would have too many edges to construct efficiently. Instead, graph building at scale requires expanding our conceptions of similarity graphs.

In this work we go beyond the standard model of a similarity graph, introducing *Stars*, a novel graph building framework for constructing extremely sparse yet high-quality similarity graphs, based on the construction of *two-hop spanners*. Unlike a classical similarity graph, two-hop spanners relax the constraint that similar points should be connected by an edge, instead requiring that such points be connected by a path of length at most two. This allows for considerable improvements in sparsity, scaling up to tens of trillions of edges, with negligible loss in quality for downstream applications.

Stars employs a specialized locality sensitive hashing technique to bucket points, and then constructs a collection of star graphs within each bucket. By creating star graphs, we significantly reduce the number of pairwise comparisons required to process a bucket from quadratic to nearly-linear in the bucket size. In practice, this results in significant (e.g. 10-20 fold) reduction in the number of comparisons made when compared to LSH alone. This reduction yields tremendous speedups for graph construction, especially for similarity measures derived from learned models, such as pre-trained deep neural networks, which are expensive to evaluate. For such learning-based models, similarity computations constitute a majority of the overall runtime, therefore reducing the number of comparisons has enormous performance benefits.

We formally introduce Stars in Section 3, and prove theoretical guarantees on its performance for similarity measures such as cosine and Jaccard similarity. For these measures, we prove that Stars can construct two-hop spanners which capture the $1/\epsilon$-approximate $k$-nearest neighbors, for any value of $k$, in time at most $\tilde{O}(n^{1+O(\epsilon)})$, for any approximation factor $(1/\epsilon) \geq 1$. Additionally, we demonstrate a variant of Stars which, given a threshold $r$, constructs a two-hop spanner such that all points with similarity larger than $r$ are connected by a path of length at most 2, and no edge connects two points with similarity less than $\epsilon r$. We demonstrate that such two-hop spanners also approximately preserve connected components, allowing for a 2-approximation of single-linkage clustering (Theorem 2.5).

We discuss implementation and design details in Section 4. We then provide an empirical evaluation of Stars in Section 5, where we demonstrate that Stars performs significantly better in practice than even the theoretical bounds suggest. Specifically, in our experiments we set $\epsilon = .99$, and demonstrate that Stars recovers $(1/\epsilon)$-approximate nearest neighbors, as well as yields $(1/\epsilon)$-approximations to threshold similarity graphs, all while constructing significantly fewer than the $O(n^{1.99})$-edges (especially for large datasets) suggested by our theoretical guarantees. In particular, Stars yields 10-20 fold and 2-10 fold improvements in number of comparisons and runtime respectively, e.g., from 120 trillion to 6 trillion comparisons, when compared to baseline LSH. The improvement is certainly much larger (at least 1000-fold) compared to the bruteforce (all-pairs) algorithm. Importantly, these runtime improvements do not negatively impact on the quality of downstream tasks, such as hierarchical clustering. The speedups become even more significant as we employ more sophisticated similarity learning models; moreover, better similarity learning models further improve the quality of the graph.

## 1.1 Related Work

**Graph Building and Nearest Neighbor Search.** The construction of similarity graphs, such as the $k$-nearest neighbor graph, is a key step for many established machine learning and data mining methods [10, 40, 11]. To efficiently compute approximate nearest neighbors, Locality Sensitive Hashing (LSH) based methods [22] and their data-dependent counterparts [4, 29, 3] have proven tremendously successful. In addition to LSH, other techniques such tree-based data structures [32, 31, 8], quantization approaches [24], and learning based methods [19], have also been utilized.

However, the majority of these methods were designed for quickly answering nearest neighbor search queries with a large dataset and a (generally) smaller number of queries, whereas in graph building the query set and the dataset are the same. This scenario allows for an additional set of techniques and optimizations to be employed, such as local search [18], and the construction of two-hop spanners (as in this work). Perhaps the most similar work to ours is [25], which builds large-scale graphs over

learned similarity models. In fact, our current results demonstrate that graphs with quality comparable to those in [25] can be constructed with substantially fewer edges via two-hop spanners.

**Large Scale Graph Clustering** Other than nearest neighbor search, one of the primary downstream analytic tasks run on similarity graphs is clustering. One of the best studied class of such algorithms is hierarchical agglomerative clustering (HAC), which has been analyzed extensively from both a theoretical and practical perspective [16, 33, 23, 38]. For graph-based HAC, it was recently shown that an approximate clustering can be obtained for average linkage in nearly linear time [17]. Previously, this was also shown for Euclidean space using Ward's linkage [1]. Several works also give sub-quadratic HAC algorithms, but sacrifice theoretical guarantees (such as approximation ratio) [15]. Finally, a related hierarchical clustering algorithm is the MST-based clustering algorithm Affinity [5], a highly-scalable method which we also use for evaluating our two-hop spanners.

**Two-Hop Spanners.** The concept of a two-hop spanner stems from the literature on metric spanners (see the surveys [21, 41, 2]), which are sparsifications of a graph with the property that distances in specifier well approximate distances in the original graph. Note that a two-hop spanner itself satisfies this property. The first usage of two-hop spanners can be attributed to [27], who demonstrated the existence of sparse two-hop spanners for Euclidean space with the $\ell_2$ metric. Two-hop spanners have also been implicitly studied in the context of min-size clustering in the MPC model [20].

## 2 Preliminaries

Let $P$ be any set of points, and let $\mu : P^2 \to \mathbb{R}$ be either a *similarity* or a *distance* measure over $P$. For the remainder, we choose to focus only on similarity measures for clarity of presentation. However, we remark that the techniques we present generalize naturally to distance measures by employing the appropriate locality sensitive hash functions for those measures (see definitions below).

We consider several types of similarity measures in the paper, including the dot-product similarity $\mu(x, y) = \langle x, y \rangle$ between points $x, y \in \mathbb{R}^d$, and the *cosine similarity* $\mu(x, y) = \cos(\theta_{x,y})$, where $\theta_{x,y}$ is the angle between the vectors $x, y$. Additionally, we consider the *Jaccard similarity* between sets: $\mu(A, B) = |A \cup B|/|A \cap B|$, where $A, B \subset \mathcal{U}$ for some universe $\mathcal{U}$, and the *weighted Jaccard similarity* between non-negative vectors $x, y$, given by by $\mu(x, y) = \frac{\sum_i \min(x_i, y_i)}{\sum_i \max(x_i, y_i)}$. Finally, we use *learned similarity measures* $\mu$ which are computed by a pre-trained model, such as a deep neural network, and may be expensive to evaluate.

**$\mu$-Nearest Neighbors.** Given $(P, \mu)$ where $n = |P|$, for any $p \in P$ we define the nearest neighbor ordering $\pi_p : [n] \to P$ as any bijection which satisfies $\mu(p, \pi_p(1)) \geq \mu(p, \pi_p(2)) \geq \ldots \geq \mu(p, \pi_p(n))$. We write $\tau_i(p) = \mu(p, \pi_p(i))$ to denote the similarity to the $i$-th nearest neighbor, and we denote the set of $i$-nearest neighbors via $N_i(p) = \{\pi_p(1), \ldots, \pi_p(i)\}$.

**Locality Sensitive Hash Families.** We now introduce the notion of a locality sensitive hash (LSH) family $\mathcal{H}$ for similarity measures. Given a similarity measure $\mu$ over a point set $P$, an LSH family $\mathcal{H}$ is a family of hash functions $h : P \to U$, for some universe $U$, with the property that $\mathbf{Pr}_{h \sim \mathcal{H}} [h(p) = h(q)]$ is larger when the points $p, q$ are more similar, namely when $\mu(p, q)$ is large. For ease of presentation, we use a simplified parameterization of LSH families, as opposed to the more standard parameterization of $(r_1, r_2, p_1, p_2)$-sensitive families (see, e.g. [26]).

**Definition 2.1 ($(r_1, r_2, \rho)$-sensitive Family)** *Fix any parameters $r_1 < r_2 \in \mathbb{R}$, and $\rho \in [0, 1]$. Let $P$ be any point set with similarity measure $\mu : P^2 \to \mathbb{R}$, and fix a universe of buckets $U$. Then a family $\mathcal{H}$ of hash functions $h : P \to U$ is said to be a $(r_1, r_2, \rho)$-sensitive locality sensitive hash family for $(P, \mu)$ if the following holds: for any $p, q \in P$, if $\mu(p, q) > r_2$, then $\mathbf{Pr}_{h \sim \mathcal{H}} [h(p) = h(q)] \geq n^{-\rho}$, and if $\mu(p, q) < r_1$ then $\mathbf{Pr}_{h \sim \mathcal{H}} [h(p) = h(q)] < n^{-4}$.*

**Graph Building and Similarity Graphs.** In this work, we consider two distinct notions of similarity graphs, which are appropriate in different contexts based on the application. The first notion is that of an $r$-threshold graph, which uniformly connects points with similarity above a given threshold.

**Definition 2.2 ($r$-threshold graph)** *Given a point set $P$, similarity measure $\mu$, and a threshold $r$, the $r$-threshold graph for $(P, \mu)$ is the graph $G = (P, E)$ where $E = \{(x, y) \in P \mid \mu(x, y) \geq r\}$.*

The $r$-threshold graph for $(P, \mu)$ is a useful similarity graph when downstream tasks impose a uniform threshold on when points should be connected. For instance, clustering with the constraint that pairs of points within a cluster have similarity above a threshold $r$. On the other hand, from the perspective of nearest neighbor search, one would like edges to be created *non-uniformly*, where a point $p$ is connected to its $k$ closest neighbors. Formally:

**Definition 2.3 ($k$-near neighbor graph ($k$-NN graph))** *Given a set of $n$ points $P$ and similarity measure $\mu$, let $\pi_p : P \to [n]$ be a nearest neighbor ordering over $P$. Then the $k$-nearest neighbor graph for $(P, \mu)$ is the graph $G = (P, E)$ where $E = \cup_{p \in P}\{(p, q) \mid q \in N_k(p)\}$.*

**Two-Hop Spanners.** A two-hop spanner is a similarity graph which relaxes the guarantees of the prior two notions of similarity graphs, where similar points are only required to be connected by a path of length 2, rather than by a direct edge. The advantage of such spanners is that they can be considerably sparser than the associated similarity graph, and still produce high-quality results for downstream tasks (such as clustering). For any graph $G = (V, E)$, $v \in V$, and integer $k \geq 1$, we write $\mathcal{N}_k(p)$ to denote the $k$-hop neighborhood of $p$: namely, the set of vertices $u \in V$ which are at (unweighted) shortest-path distance at most $k$ from $v$ in $G$. The following definition captures the generalization of $r$-threshold graphs from one-hop to two-hop neighborhoods.

**Definition 2.4 (Threshold Two-Hop Spanners)** *Given a point set $P$ and similarity measure $\mu$, a $(r_1, r_2)$-two-hop spanner is a graph $G = (P, E)$, such that: (1) for every edge $(p, q) \in E$, we have $\mu(p, q) \geq r_1$, and (2) for every pair $p, q \in P$ with $\mu(p, q) \geq r_2$, we have $p \in \mathcal{N}_2(q)$.*

The above notion of two hop-spanners provide a good (two-hop) approximation of the $r$-threshold graph for a point set. The approximation is controlled by the gap between $r_1, r_2$. In Section 3.1, we give provable guarantees for Stars, by demonstrating that it constructs a two-hop spanners where this gap is small. In Section 3.2, we define an analogous notion of two-hop spanners for approximation the $k$-nearest neighbor graph, and prove that Stars efficiently constructs such an approximation for $k$-nearest neighbors as well.

Two-hop spanners can be used as a powerful approximation of the underlying similarity structure of dataset. For instance, we demonstrate that two-hop spanners can be utilized to obtain an approximation to the optimal $k$-single linkage clustering solution, which is to partition $P$ into $k$ clusters $C_1, \ldots, C_k$ so as to minimize $\max_{i \neq j} \max_{p \in C_i, q \in C_j} \mu(p, q)$.

**Theorem 2.5** *Let $c \geq 1$ and $r < OPT_k/c$, where $OPT_k$ is optimal cost of $k$-single-linkage clustering on $(P, \mu)$. Then any $(r/c, r)$-2-hop spanner $G$ has at least $k$ connected components. Furthermore, for any two connected components $C, C'$ of $G$, $\min_{x \in C, y \in C'} \mu(x, y) \geq r$. Thus, by constructing $\log\left(\frac{\max_{x,y \in p} \mu(x,y)}{\min_{x,y \in P} \mu(x,y)}\right)$ distinct $(r/c, r)$-2-hop spanner for geometrically increasing $r$, one obtains a 2-approximation to $k$-single-linkage clustering.*

## 3 Graph Building via Two-Hop Spanners

In this section, we theoretically analyze the performance of the Stars algorithm, by demonstrating that it efficiently constructs highly-sparse two-hop spanners. We describe two variants of Stars, in Sections 3.1 and 3.2 respectively. The first can be used to approximate the $r$-threshold similarity graph, and the second can be used to approximate the $k$-nearest neighbor graph.

### 3.1 Two-Hop Spanners via Locality Sensitive Hashing

We first describe the Stars algorithm for constructing a two-hop spanner for approximating $r$-similarity threshold graph for a point set $P$ and similarity measure $\mu$. Specifically, given a locality sensitive hash family $\mathcal{H}$ for $(P, \mu)$, Stars proceeds by bucketing points via a randomly drawn hash function $h \sim \mathcal{H}$. Within each bucket, Stars samples a random "leader" $p$, and then connects together all points in the bucket to $p$ which are sufficiently similar to $p$, effectively creating a star graph centered at $p$. The process is repeated with $R$ independent sketches, to ensure that all sufficiently similar points land in the same bucket at least once. The formal algorithm is given in the algorithm Stars 1.

**Theorem 3.1** *Let $P$ be a point set equipped with a similarity measure $\mu$, and fix any $r_1 < r_2 \in \mathbb{R}$. Let $\mathcal{H} : P \to U$ be a $(r_1, r_2, \rho)$ family of hash functions, such that each $h \in \mathcal{H}$ can be evaluated*

---

**Stars 1: Constructing Approximate Threshold Graphs**

---

**Input:** Point set $P$, similarity measure $\mu$, and a $(r_1, r_2, \rho)$-sensitive hash family $\mathcal{H}$.

**Repeat:** the following procedure $R = c_1 n^\rho \log n$ times, for a sufficiently large constant $c_1$.

1. Evaluate $h(p)$ for each $p \in P$, and construct the buckets $B_u = \{p \in P \mid h(p) = u\}$.

2. For each bucket $B_u$, where $u \in U$:

   - Sample a uniformly random leader $x \sim B_u$.
   - For all $y \in B_u \setminus \{x\}$, if $\mu(x, y) > r_1$, then create an edge $(x, y)$ with weight $\mu(x, y)$.

---

in time at most $Time(\mathcal{H})$. Let $G = (P, E)$ be the graph produced by the algorithm Stars 1, using the LSH family $\mathcal{H}$. Then with probability $1 - 1/poly(n)$, $G$ is a $(r_1, r_2)$-two-hop spanner for $(P, \mu)$. Moreover, the number of edges produced is at most $|E(G)| = \tilde{O}(n^{1+O(\rho)})$, and the runtime of the algorithm is bounded by $\tilde{O}(n^{1+O(\rho)} \cdot Time(\mathcal{H}))$.

In the Appendix, we demonstrate that for parameters $\alpha, \epsilon > 0$, there exist $(1 - \epsilon^{-1}\alpha, 1 - \alpha, O(\epsilon))$-sensitive hash families for both cosine and Jaccard similarity measures.

## 3.2 Two-Hop Spanners via SortingLSH

We now describe a variant of the Stars algorithm for the construction of two-hop spanners which approximate that $k$-nearest neighbor graph ($k$-NN graph, Definition 2.3). While the $k$-NN Graph itself is sparse for small $k$, for large $k$ this is no longer the case, and directly approximating the $k$-NN graph becomes infeasible. Instead, by employing two-hop spanners, we show how one can construct extremely sparse graphs, with a almost-linear number of edges, such that the approximate $k$-nearest neighbors of every point $p$ are contained in the two-hop neighborhood of $p$. To do so, we will need to utilize a technique known as SortingLSH.

**SortingLSH.** Since the similarities $\tau_k(p)$ between points $p$ and their $k$ nearest neighbors may vary significantly across the dataset, we cannot apply a single level of bucketing based on a hash family chosen with a fixed set of parameters, as doing so would result in points being split around a uniform threshold. Instead, we utilize a technique known as *SortingLSH*, which originates from the work [6]. SortingLSH evaluates a sequence of hash functions $H(p) = (h_1(p), \ldots, h_\ell(p))$ for each point $p \in P$, and interprets the resulting string of buckets as an key for $p$. One then sorts the keys $H(p)$ lexicographically, and breaks the sorted sequence into contiguous chunks of a given window size $W$. We then apply the two-hop spanner technique on each chunk, by sampling $s$ leaders within that bucket, and comparing each leader to the rest of the chunk.

The key advantage of SortingLSH is that points $p$ living in dense regions of the similarity space $(P, \mu)$ (i.e., the similarity $\tau_k(p)$ is large) will share a longer prefix of hashes with their $k$-nearest neighbors, and therefore be more likely to be placed in the same window as them. At the same time, points $p$ whose $k$-nearest neighbors are not as similar will still be equally as likely to share a (shorter) prefix with their neighbors, and therefore also placed in the same window.

In order to describe our results, we first introduce a notion of $k$-approximate nearest neighbors ($k$-ANN), defined with respect to a family of hash functions $\mathcal{H}$. In what follows, recall that given any $p \in P$ and $i \in [n]$ we write $\pi_p(i)$ to denote the $i$-th nearest neighbor to $p$ with respects to $\mu$.

**Definition 3.2 (ANN with respects to an LSH Family)** *Fix a point set $P$ equipped with a similarity measure $\mu$, and let $\mathcal{H} = \{h : P \to U\}$ be a family of hash functions. Fix any $\rho \in [0, 1]$. We say that a collection of sets $\mathcal{A} = \{\mathcal{A}_p\}_{p \in P}$ is an $(k, \rho)$-ANN family with respect to $\mathcal{H}$ if for every $p \in P$:*

- *$\mathcal{A}_p$ is a prefix of the sequence $\pi_p(1), \pi_p(2), \ldots, \pi_p(n)$ with size $|\mathcal{A}_p| \geq k$.*

- *There exists an integer $\ell = \ell(p)$ such that for all $i \in [k]$, we have:*

$$\mathbf{Pr}\left[(h_1(p), \ldots, h_\ell(p)) = (h_1(\pi(i)), \ldots, h_\ell(\pi(i)))\right] \geq n^{-\rho}$$

*and such that for all $x \notin \mathcal{A}_p$ we have*

$$\mathbf{Pr}\left[(h_1(p), \ldots, h_\ell(p)) = (h_1(x), \ldots, h_\ell(x))\right] \leq n^{-4}$$

*If $\mathcal{A}$ has the property that $\ell(p) \leq M$ for all $p \in P$, then we say that $\mathcal{A}$ is $M$-bounded $(\rho, M)$-ANN family (hereafter a $(k, \rho, M)$-ANN family).*

Definition 3.2, while at first somewhat unwieldy, is fairly straightforward to unpack. Specifically, given a base hash family $\mathcal{H}$, and let $\mathcal{H}^\ell$ denote the family of $\ell$-wise concatenations of $h \in \mathcal{H}$. Then if, for every $p \in P$, there exists a smaller threshold $r_p \leq \tau_k(p)$ and a sketch length $\ell(p)$ such that the family $\mathcal{H}^{\ell(p)}$ is a $(r_p, \tau_k(p), \rho)$-sensitive hash family (Definition 2.1), then the family $\mathcal{A} = \{\mathcal{A}_p\}_{p \in P}$, where $\mathcal{A}_p = \{q \in P \mid \mu(p, q) \geq r_p\}$, is a $(k, \rho)$-ANN family. Note that the gap between $r_p$ and $\tau_k(p)$ will depend on the properties of the family $\mathcal{H}$. Lastly, the condition of being $M$-bounded simply ensures that one can reach the appropriate splitting point $\ell(p)$ with a limited number of hash functions. Thus, for points $p$ with extremely large $\tau_k(p)$ (e.g.., near-duplicate neighbors), in order to ensure that $\mathcal{A}$ is $M$-bounded, one may need to limit $\ell(p) = M$ which would result in a value of $r_p$ that is smaller than what would otherwise be possible with a larger sketch length $\ell(p)$.

We now prove explicit bounds for the above notion of $k$-ANNs for the case of angular (cosine) similarity, and the Jaccard similarity measure. The bounds utilize the well-known SimHash family [14] for cosine similarity, and MinHash [12] for Jaccard similarity. For the case of weighted Jaccard similarity, we remark that there is a straightforward reduction (i.e. similarity preserving mapping) from the (integer) weighted to unweighted case by simply duplicating coordinates, therefore the following bounds also apply to the weighted variant. For weighted Jaccard distance over general non-integer vectors, one can instead use the variant of min-hash for probability distributions of [34].

**Proposition 3.3 (Approximate Nearest Neighbors for Angular and Jaccard Similarity)** *Let $P$ be a subset of either $\mathbb{R}^d$ or $2^{\mathcal{U}}$ for some universe $\mathcal{U}$. In the first case, let $\mu$ be the Angular Similarity $\mu(x, y) = 1 - \theta_{x,y}$ where $\theta_{x,y}$ is the angle between $x, y \in P$ (normalized so that $\theta_{x,y} \in [0, 1]$), and in the second case we set $\mu$ to be the Jaccard similarity $\mu(A, B) = \frac{|A \cap B|}{|A \cup B|}$ between sets $A, B \subset \mathcal{U}$. For any $p \in P$, let $s_k(p) = 1 - \tau_k(p)$. Then for any $\epsilon \in (0, 1)$ and $M \geq 1$, there exists a hash family $\mathcal{H}$ for $(P, \mu)$ such the family $\mathcal{A} = \{\mathcal{A}_p\}_{p \in P}$, where*

$$\mathcal{A}_p = \left\{x \in P \mid \mu(p, x) \geq \min\left\{1 - \epsilon^{-1}s_k(p), \ 1 - 1/M\right\}\right\}$$

*is a $(k, O(\epsilon), 4M \log n)$-ANN family with respect to $\mathcal{H}$.*

Since both the angular and Jaccard similarity are bounded by 1, the value $s_k(p)$ in the Proposition 3.3 can be interpreted as a natural *dissimilarity* measure $d(p, q) = 1 - \mu(p, q)$. Therefore, Proposition 3.3 demonstrates that we can take $\mathcal{A}_p$ to be the set of points $q$ which are at most a $1/\epsilon$ factor farther away than the $k$-th nearest neighbor to $p$, namely $\mu(p, q) \leq \epsilon^{-1}d(p, \pi_k(p))$.

We now state the main result of this section, which demonstrates that Stars provably recovers nearly $k$ distinct $(1/\epsilon)$-approximate $k$-nearest neighbors in the two-hop neighborhood of $p$. Formally, for each point $p \in P$, we guarantee that there is a two-hop path from $p$ to nearly $k$ points $q \in \mathcal{A}_p$. Moreover, we guarantee that this path utilizes only edges between other approximate nearest neighbors $u \in \mathcal{A}_p$.

**Theorem 3.4** *Let $P$ be a point set equipped with a similarity measure $\mu$, and fix any $\delta \in (0, 1)$. Let $\mathcal{H} : P \to U$ be a family of hash functions, such that each $h \in \mathcal{H}$ can be evaluated in time at most $Time(\mathcal{H})$. Let $G = (P, E)$ be the graph produced by the algorithm Stars 2. Let $\mathcal{A} = \{\mathcal{A}_p\}_{p \in P}$ be a $(k, \rho, M)$-ANN family with respects to $\mathcal{H}$. For any $p \in P$, let $\hat{G}_p$ denote the subgraph of $G$ induced by the set $\mathcal{A}_p \cup \{p\}$. Then with probability $1 - 1/poly(n)$, for every $p \in P$, we have: $|\mathcal{N}^2_{\hat{G}_p}(p)| \geq (1 - \delta)k$. Moreover, we have $|E(G)| = \tilde{O}(\delta^{-1}n^{1+O(\rho)})$, and the runtime of the algorithm is $\tilde{O}(\delta^{-1}n^{1+O(\rho)}M \cdot Time(\mathcal{H}))$.*

Theorem 3.4 demonstrates that, even for extremely large values of $k$, for instance $k = \sqrt{n}$, one can construct a two-hop spanner with a nearly-linear number of edges, such that each point $p$ is connected with at least $(1 - \delta)k$ of its $k$-approximate nearest neighbors, via a path containing only $k$-approximate nearest neighbors of $p$. Notably, the runtime and the size of the graph produced are *independent* of $k$.

---

**Stars 2: Constructing Approximate Nearest Neighbor Graphs**

---

**Input:** Point set $P$, a hash family $\mathcal{H}$, ANN parameters $k, \rho, M > 0$, recall parameter $\delta \in (0, 1)$.

**Repeat:** the following procedure $R = c_1 n^\rho \log n$ times, for a sufficiently large constant $c_1$.

1. Let $(h_1, h_2, \cdots, h_M)$ be a sequence of independent draws from $\mathcal{H}$.
2. Sort the points $x$ lexicographically according to the hash values $(h_1(x), \ldots, h_M(x))$. Let $x_1, \ldots, x_n$ be this ordering. Set the window size to be $W = 16k$
3. Pick a random shift $r \sim [W/2, \ldots, W]$, and split $x_1, \ldots, x_n$ into consecutive blocks $B_1, B_2, \ldots, B_\ell$, each of size at most $W$, such that $B_1 = \{x_1, \ldots, x_r\}$.
4. **If:** $k > n^{2\rho}$
   (a) For each block $B_j$, sample $s = c_2 \delta^{-1} n^\rho \log^2 n$ uniformly random leaders $c_2$ $y_1^j, \ldots, y_s^j \sim B_j$, where $c_2$ is a sufficiently large constant depending only on $c_1$.
   (b) Create an edge $(x, y_i^j)$ with weight $\mu(x, y_i^j)$ for every $x \in B_j$ and leader $y_i^j \in B_j$
5. **Else if:** $k \le n^{2\rho}$
   (a) For each block $B_j$, create an edge $(x, y)$ with weight $\mu(x, y)$ for every pair $x, y \in B_j$.

---

## 4 System Implementation and Design

We have implemented Stars as part of the Grale [25] graph building system using Flume - a C++ counterpart to FlumeJava [13]. which is based on the Adaptive Massively Parallel Computation (AMPC) model [7]. Each logical unit of computation is automatically distributed across a number of worker machines, with the experiments in this paper scaling to thousands of individual workers.

Our implementation has two primary phases: generating LSH tables using LSH or SortingLSH, then scoring pairs of points that share a sketch using all-pairs or Stars. For efficiency we generate LSH tables containing only the identifier of each point, excluding the associated point features, as the same point is sketched $R > 1$ times to aid in edge recall. To compute pairwise similarity we thus require an additional round of communication join point features with the LSH tables. We implement this in one of two ways: a MapReduce-style distributed shuffle sort, or via lookups in a distributed hash table (DHT). These two options trade off CPU time and disk usage for memory.

In the shuffle case, we require $O(Rn)$ additional disk storage and $O(Rn \log(Rn))$ time to materialize the joined table. For problem settings with billions of points and large feature vectors, this extra storage requirement can be prohibitive. The DHT caches the entire input dataset in memory across multiple machines, requiring $O(n)$ RAM but no additional on-disk storage. This enables online feature lookup as we process each bucket, eliminating the need for a shuffle and costly disk I/O.

An additional important implementation detail is the limit we impose on LSH bucket sizes. In the worst case, a naive LSH implementation could have the same or worse running time than a brute-force all pairs comparison, because a poorly chosen LSH function could hash the entire dataset to a single value, up to $R$ times. To ensure robustness to sub-optimal LSH settings, we randomly partition large buckets into size-constrained sub-buckets prior to pairwise scoring. This reduces the overall number of edges eligible for comparison, potentially impacting edge recall, but also capping the worst case running time for the scoring phase of graph construction. Due to its nearly-linear runtime complexity, the Stars algorithm enables us to relax the sub-bucket size limitation significantly.

See Appendix C for further implementation details.

## 5 Empirical Study

We evaluate the performance and quality of Stars on both real and synthetic datasets of varying scales. We compare the LSH+Stars, SortingLSH+Stars, SortingLSH, and bruteforce (AllPair) algorithms.

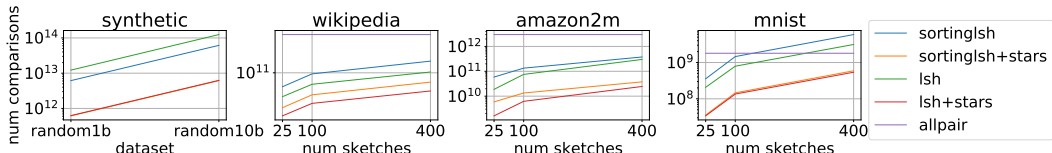

Figure 1: Number of comparisons of each algorithm on each dataset. For Random1B and Random10B, we only run algorithms with number of sketches $R = 25$, and the AllPair algorithm does not finish in 3 days.

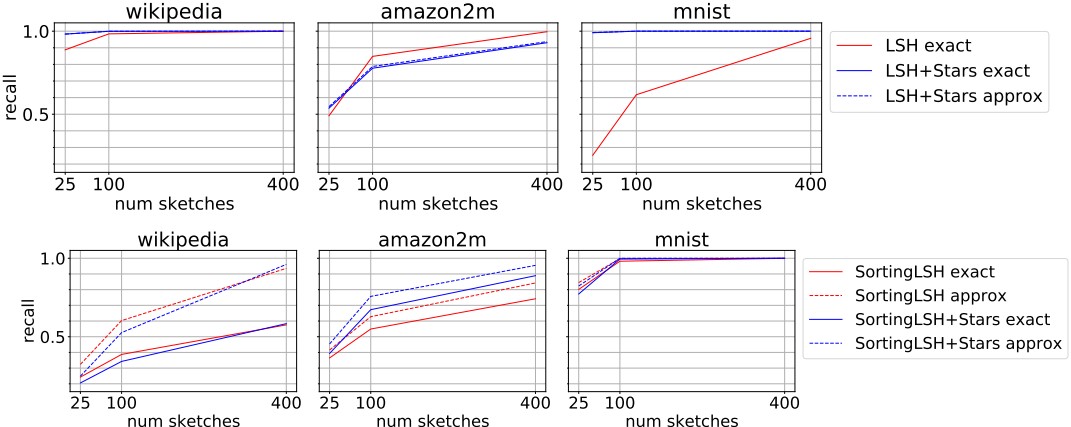

Figure 2: The recall of found near(est) neighbors of each algorithm.

**Datasets.** We run experiments on three real public datasets: MNIST [30], Amazon2m [9] (also known as OGBN-Products [28]), and Wikipedia [39], and two synthetic datasets: random1B and random10B. MNIST contains 60k data points, each of which has a feature of a 784-dimensional float vector. Wikipedia contains 3,650,339 data points, where each point is represented by a set of strings with positive weights. Amazon2m contains 2,449,029 data points, each of which has a feature of a 100-dimensional float vector and a set of strings. Random1B and random10B are generated from a Gaussian mixture model with 100 modes, where each data point has 100 dimensions. Random1B contains $10^9$ data points and Random10B contains $10^{10}$ data points. We refer readers to Appendix D for further details regarding the datasets.

**Sketching parameters.** For each similarity measure studied, we use a corresponding LSH function to build the graph. In particular, for MNIST, Random1B and Random10B datasets, we study the cosine similarity between float vector features, and thus we employ SimHash for them. For the Wikipedia dataset, we study the weighted Jaccard similarity between two sets of strings with weights, and therefore we use the weighted Minhash LSH to build the graph. For the Amazon2m dataset, we study two different similarity measures: (1) a mixture of cosine similarity and Jaccard similarity, and (2) a neural network where the training set of candidate pairs are generated by SimHash over float vector features and MinHash over sets of strings [25]. In summary, in both cases, we use a mixture of SimHash and MinHash for graph building. See Appendix D for more detailed sketching setups.

**Number of comparisons.** In Figure 1, we illustrate the number of pairwise similarity comparisons of each algorithm on each dataset. In all cases, Stars yields at least a $\sim 10$-fold improvement in number of comparisons over the other algorithms. The number of comparisons used by Stars can be further reduced by choosing a smaller number of leaders (see Appendix D).

**Coverage of Near(est) Neighbors.** We evaluate the number of (approximate) near(est) neighbors which can be found for each point in the dataset, and by each algorithm. We run the bruteforce (AllPair) algorithm for MNIST, Wikipedia and Amazon2m (using mixture similarity) datasets to find all ground truth 100-nearest neighbors and all ground truth near neighbor points with similarity above $0.5$ for each point. For the non-Stars LSH based algorithm, we compute the fraction of points with similarity at least $0.5$ that are direct neighbors for each point. For LSH+Stars, we compute two ratios, the first is the fraction of points with similarity at least $0.5$ that can be found in two hops where each edge has similarity also at least $0.5$, and the second is the same except that the two hop edges

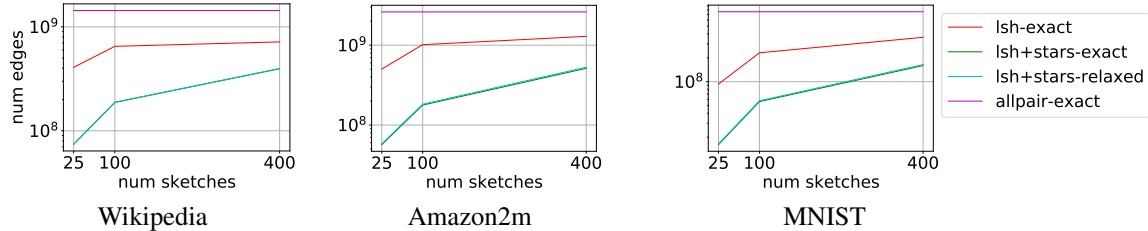

Figure 3: The number of edges with similarity at least 0.5 (0.495 for relaxed threshold) built by each algorithm.

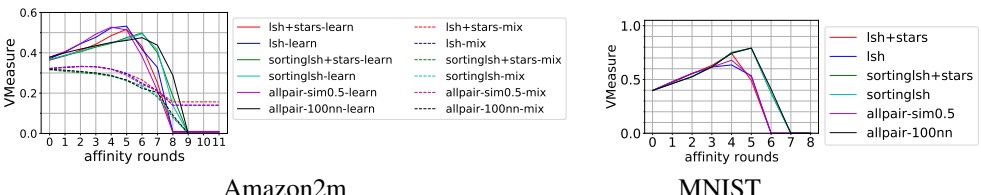

Figure 4: The VMeasure scores of clusterings. The *allpair-100nn* indicates the ground truth 100-nearest neighbor graph. The *allpair-sim0.5* indicates the ground truth near neighbor graph induced by all edges with similarity at least 0.5. The suffix *learn* indicates the similarity learned by neural network. The suffix *mix* indicates the mixture similarity of cosine similarity and Jaccard similarity.

can have similarity at least $0.495$ (this aligns with the $1.01$-approximation mentioned in Section 3.2). For SortingLSH based algorithms, we consider the fraction of exact $100$-nearest neighbors can be found in one hop and two hops for non-Stars and Stars respectively in the graph introduced by the $100$-nearest neighbors. We also consider the relaxation of finding $1.01$-approximate $100$-nearest neighbors (i.e. $1/\epsilon = 1.01$). Note that if we can find more than $100$ approximate $100$-nearest neighbors, we regard the ratio as $1$. In Figure 2, we report the average of each ratio over all data points. The graphs built using Stars are able to find more near(est) neighbors in 2 hops, with fewer edges overall. Note that the graphs built by SortingLSH-based algorithms have the same sparsity since we only keep the $250$ closest points for each node (even for SortingLSH+Stars). The sparsity of graphs built by LSH-based algorithms is presented in Figure 3. In Figure 2, we show a good recall of finding $1.01$-approximate near(est) neighbors in 2 hops by our Stars algorithms. In the meanwhile, the number of edges shown in Figure 3 is much less than $n^{1.99}$ which is suggested by our theoretical guarantees. This phenomenon is observed in all 3 datasets used in Figure 2,3.

**Clustering.** The points from MNIST are drawn from $10$ classes, and the points from Amazon2m are drawn from $47$ classes. To cluster the graphs, we run average Affinity clustering [5] on the graphs built by each algorithm. In particular, for graphs built by LSH-based algorithms, we only keep the edges with similarity at least $0.5$, and for graphs built by SortingLSH-based algorithms, we only keep the closest direct $100$ nodes for each point. For Amazon2m, we consider both mixture similarity and learning similarity, i.e., we built graphs for each similarity and apply clustering on these graphs. For all algorithms, we use the number of sketches $R = 400$. We measure the clustering quality via VMeasure score [37] which is the harmonic mean between homogeneity score and completeness score of a clustering. The VMeasure score is in $(0, 1)$. The higher the VMeasure score, the higher the quality of the clusters. We report VMeasure scores in Figure 4.

**Effect of the similarity function.** As shown in Figure 4, using a similarity function learned by a neural network in the graph building stage indeed helps other downstream tasks such as clustering on the built graphs. However, a sophisticated similarity function increases the running time of computing the similarity between two points. In fact, all of our algorithms become $5\times \sim 10\times$ (in terms of the total running time over all workers) slower when using neural network similarity instead of using the mixture of cosine and Jaccard similarity to build the graph for Amazon2m dataset. Fortunately, the Stars graph building algorithms use significantly fewer comparisons, and are $10\times$ faster than the non-Stars versions (in terms of total running time over all workers). In addition, they lose only negligible quality for downstream tasks. Thus, given the same computational resources, switching to a Stars-based graph building strategy enables us to employ a wider range of similarity functions, and affords us the potential to significantly improve the quality of downstream tasks.

**Experiments on Random10B dataset.** Since we set the degree threshold to be 250, each algorithm outputs exactly 2.5 trillion edges for the Random10B dataset. However, since the non-Stars-based algorithms make $\sim 10^{14}$ comparisons, more than 95% of these comparisons are redundant. In contrast, the number of comparisons made by Stars-based algorithms is only 2-3$\times$ the number of edges obtained. Overall, the total running time of non-Stars LSH based algorithm is 100$\times$ the total running time of LSH+Stars. Similarly, the total running time of non-Stars SortingLSH algorithm is 10$\times$ the total running time of SortingLSH+Stars. The actual runtime of all Stars-based algorithms finish in 2 hours, whereas the non-Stars SortingLSH algorithm finishes in 4 hours, and the non-Stars LSH algorithm finishes in 23 hours.

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
