## Appendix

## A Connected Components and Single-linkage Clustering

We now demonstrate that given a $(r_1, r_2)$-two-hop spanner for $(P, \mu)$, one can obtain an approximate solutions to single-linkage clustering by varing $(r_1, r_2)$.

**Observation A.1** *For $c \geq 1$ and $r > 0$, if two points are in the same connected components of $r$-threshold graph, they must be in the same connected components of $(r/c, r)$-two-hop spanner. Furthermore, if two points are in the same connected components of $(r/c, r)$-two-hop spanner, they must be in the same connected components of $r/c$-threshold graph.*

**Corollary A.2** *For $c \geq 1$ and $r > 0$, the number of connected components of $(r/c, r)$-two-hop spanner is at least the number of connected components of $r/c$-threshold graph, and is at most the number of connected components of $r$-threshold graph.*

Given a parameter $k \geq 1$, the goal of the single-linkage clustering is to partition the input data $P$ into $k$ clusters $C_1, C_2, \cdots, C_k$ such that the maximum similarity between points in separate clusters, namely the quantity minimized $\max_{i \neq j} \max_{p \in C_i, q \in C_j} \mu(p, q)$ is minimized Let

$$OPT_k = \min_{C_1, C_2, \cdots, C_k} \max_{i \neq j} \max_{p \in C_i, q \in C_j} \mu(p, q)$$

i.e., $OPT_k$ denotes the optimal cost of $k$-single-linkage clustering. Let $r$ be in the range $[OPT_k, OPT_{k+1})$. Then by the definition of $OPT$, it is easy to verify that the connected components of $r$-threshold graph yield the optimal $k$-single-linkage clustering. In the following, we show that we can obtain an approximate $k$-single-linkage clustering via a two-hop spanner with the appropriate parameters.

**Theorem A.3** *Let $c \geq 1$ and $r < OPT_k/c$, where $OPT_k$ is optimal cost of $k$-single-linkage clustering on $(P, \mu)$. Then any $(r/c, r)$-2-hop spanner $G$ has at least $k$ connected components. Furthermore, for any two connected components $C, C'$ of $G$, $\min_{x \in C, y \in C'} \mu(x, y) \geq r$, and the number of connected components of $G$ is at least $k$.*

**Proof:** By the definition of $(r/c, r)$-2-hop spanner, if $\mu(p, q) > r$, there is no edge between $p$ and $q$. Let us show that the connected components of $G$ is at least $k$. We prove it by contradiction. Let $C_1, C_2, \cdots, C_k$ be the optimal $k$-single-linkage clusters. Suppose $G$ has at most $k - 1$ clusters. By pigeonhole principle, we can find $p, q$ which are in the same connected components but $p \in C_i, q \in C_j$ for some $i \neq j$. Since $p, q$ are in the same connected components, there must be a path between $p$ and $q$ in $G$. We can find an edge $(p', q')$ on the path such that $p' \in C'_{i'}, q' \in C'_{j'}$ for some $i' \neq j'$. Since $C_1, C_2, \cdots, C_k$ is the optimal $k$-single-linkage clustering, we have $\mu(p', q') \geq OPT_k > c\dot{r}$ which leads to a contradiction.

Next let us show that for any two connected components $C, C'$ of $G$, $\min_{x \in C, y \in C'} \mu(x, y) > r$. Suppose there are two points $p, q$ satisfying $\mu(p, q) \leq r$. By the definition of 2-hop spanner, there is a path between $p$ and $q$ with at most 2 hops. Therefore, for any two points that are in different connected components, their distance is at least $r$. $\square$

A simple corollary of the above theorem is that we can easily obtain a $k$-single-linkage clustering solution with cost at least $r$ by arbitrarily merging connected components to reduce the number of clusters to $k$.

## B Proofs Omitted from Section 3

In this section, we fill in missing details and proofs from statements and theorems made iun Section 3. Firstly, we prove Theorem 3.1 below. Afterwards, we first formalize the discussion on the existence of certain $(r_1, r_2, \rho)$-sensitive families for the Jaccard and Cosine similarity measures.

### B.1 Proof of Theorem 3.1

**Proof:** First, note that by construction, the algorithm Stars 1 never creates an edge between two points $x, y \in P$ with $\mu(x, y) < r_1$. Thus, the first condition for a $(r_1, r_2)$-two hop spanner (Definition

2.4) is satisfied deterministically. To show the second condition, fix any point $p \in P$, and fix any $q \in P$ with $\mu(p, q) > r_2$. We show that $q \in \mathcal{N}_2(p)$ with high probability, which will complete the proof.

Firstly, by definition of a $(r_1, r_2, \rho)$-sensitive family, the probability that $p, z$ collide in a hash bucket $B_u$ when $\mu(p, z) < r_1$ is at most $1/n^4$. Thus, the probability that any pair $p, z$ collide with $\mu(p, z) < r_1$ in a single repetition is at most $1/n^2$, and by a union bound at most $R/n^2 < 1/(2n)$ over all $R = c_1 n^\rho \log n$ repetitions, where $c_1$ is a sufficiently large constant. Call this event $\mathcal{E}$, and condition on it now. Then, again by definition of a $(r_1, r_2, \rho)$-sensitive family, we have that $p, q$ collide in a hash bucket $B_u$ with probability at least $n^{-\rho}$. Thus, by repeating the hashing $R = c_1 n^\rho \log n$ times, it follows $p, q$ collide in at least one repetition with probability at least $1 - 1/n^4$. In this repetition, by event $\mathcal{E}$, we have that all pairs of points which are contained in $B_u$ have pairwise similarity at least $r_1$. In particular, if $x \sim B_u$ is the uniformly sampled leader in algorithm Stars 1, we have that $\mu(x, p) > r_1$ and $\mu(x, q) > r_1$, thus the edges $(x, p), (x, q)$ will be added to the graph with probability at least $1 - \Pr[\mathcal{E}] - 1/n^{-4} > 1 - 1/n$, which completes the proof. $\qquad\square$

We now formalize the claim made after the statement of Theorem 3.1 about the existence of $(r_1, r_2, \rho)$-sensitive families for the Jaccard and cosine (angular) similarity measures, where the latter is given by $\mu(x, y) = 1 - \theta_{x,y}$. The proof of the following proposition follows immedietly from the proof of Proposition 3.3, by simply replacing the threshold $\theta_k(p)$ with the fixed threshold $\alpha$, and setting $M = \infty$ (to avoid the second case in Proposition 3.3, which is not needed for the following claim).

**Proposition B.1** *Let $\mu$ be either the angular or Jaccard similarity measure on a dataset $P$. Fix any $\epsilon, \alpha \in (0, 1)$. Then there exists a $(1 - \epsilon^{-1}\alpha, 1 - \alpha, O(\epsilon))$-sensitive hash family for $\mu$.*

### B.2   Proof of Propositions 3.3

We now provide the proof of Proposition 3.3.

**Proposition B.2 (Approximate Nearest Neighbors for Angular Similarity)** *Let $P \subset \mathbb{R}^d$ be a subset, and let $\mu(x, y) = 1 - \theta_{x,y}$ where $\theta_{x,y}$ is the angle between $x, y \in P$ (normalized so that $\theta_{x,y} \in [0, 1]$). Let $\mathcal{H}$ be the SimHash family, where $h \sim \mathcal{H}$ is selected by first drawing $z \in \mathbb{R}^n$ uniformly from the unit sphere, and setting $h(x) = \text{SIGN}(\langle x, z \rangle)$. Then for any $\epsilon \in (0, 1)$, integer $M \geq 1$, and $p \in P$, let $\theta_k(p) = 1 - \tau_k(p)$ be the normalized $k$-th closest angle to the point $p$. Then the family $\mathcal{A} = \{\mathcal{A}_p\}_{p \in P}$, where*

$$\mathcal{A}_p = \left\{ x \in P \mid \mu(p, x) \geq \min\left\{ 1 - \frac{\theta_k(p)}{\epsilon}, 1 - \frac{1}{M} \right\} \right\}$$

*is a $(k, O(\epsilon), 4M \log n)$-ANN family with respect to $\mathcal{H}$.*

**Proof:**  Fix any $p \in P$, and set $\ell = \ell_p = \min\{\frac{4\epsilon \log n}{\theta_k(p)}, 4M \log n\}$. In the first case, suppose $\ell = \frac{4\epsilon \log n}{\theta_k(p)} \leq 4M \log n$. Then note that, for any $x \in N_k(p)$, the probability that $(h_1(p), \ldots, h_\ell(p)) = (h_1(x), \ldots, h_\ell(x))$ is at least

$$(1 - \theta_k(p))^\ell = (1 - \theta_k(p))^{\frac{4\epsilon \log n}{\theta_k(p)}} \geq n^{-O(\epsilon)}$$

Similarly, for $x \notin \mathcal{A}_p$, we have $\mu(p, x) < 1 - \frac{\theta_k(p)}{\epsilon}$, so the probability that $(h_1(p), \ldots, h_\ell(p)) = (h_1(x), \ldots, h_\ell(x))$ is at most

$$\left(1 - \frac{\theta_k(p)}{\epsilon}\right)^{\frac{4\epsilon \log n}{\theta_k(p)}} \leq 1/n^4$$

where we used the inequality that $(1 - x)^{n/x} \leq \left(\frac{1}{2}\right)^n$ for any $x \in (0, 1]$ and $n \geq 1$.

Next, suppose that $\ell_p = 4M \log n < \frac{4\epsilon \log n}{\theta_k(p)}$. In this case, since $\ell_p$ is only smaller than the threshold used above, we still have that $h_1(p), \ldots, h_\ell(p) = h_1(x), \ldots, h_\ell(x)$ with probability at least $n^{-O(\epsilon)}$ for any $x \in N_k(p)$. Next, for any $x \notin \mathcal{A}_p$, the probability that $(h_1(p), \ldots, h_\ell(p)) = (h_1(x), \ldots, h_\ell(x))$ is at most

$$\left(1 - \frac{1}{M}\right)^{\ell_p} = \left(1 - \frac{1}{M}\right)^{4M \log n} \leq 1/n^4$$

which completes the proof.

$\square$

**Proposition B.3 (Approximate Nearest Neighbors for Jaccard Similarity)** *Let $P \subset 2^{\mathcal{U}}$ be a set of subsets of a universe $\mathcal{U}$, and let $\mu(x, y) = \frac{|x \cap y|}{|x \cup y|}$ be the Jaccard Similarity between $x, y \subset \mathcal{U}$. Let $\mathcal{H}$ be the MinHash family, where $h \sim \mathcal{H}$ is selected by first drawing a random number $n_u \sim [0, 1]$ for each $u \in \mathcal{H}$, and setting $h(x) = \min_{u \in x} n_u$. Then for any $\epsilon \in (0, 1)$, integer $M \geq 1$, and $p \in P$, let $s_k(p) = 1 - \tau_k(p)$ be the $k$-th smallest Jaccard distance to the point $p$. Then the family $\mathcal{A} = \{\mathcal{A}_p\}_{p \in P}$, where*

$$\mathcal{A}_p = \left\{ x \in P \mid \mu(p, x) \geq \min \left\{ 1 - \frac{s_k(p)}{\epsilon}, 1 - \frac{1}{M} \right\} \right\}$$

*is a $(k, O(\epsilon), 4M \log n)$-ANN family with respect to $\mathcal{H}$.*

**Proof:** The proof of Proposition B.3 is identical to that of Proposition B.2, by simply using that fact that $\Pr[h(x) = h(y)] = \frac{|x \cap y|}{|x \cup y|}$ for the case of MinHash. $\square$

## B.3 Proof of Theorem 3.4

**Proof:** We begin by demonstrating that the first condition holds, by demonstrating that our algorithm finds enough edges so that $\left| \mathcal{N}_{\hat{G}_p}^2(p) \right| \geq (1 - \delta)k$. In what follows, fix any vertex $v \in P$. We run Stars 2 with number of iterations $R = O(\log n \cdot n^\rho)$. For each $i \in [R]$, let $H_i = (h_{i,1}, h_{i,2}, \ldots, h_{i,M})$ be the hash functions drawn on repetition $i$, and for $t \in [M]$, let $H_i^t = (h_{i,1}, h_{i,2}, \ldots, h_{i,t})$ be the first $t$ hash functions drawn in repetition $i$.

For each repetition $i \in [R]$, call $i$ balanced for the point $p$ if $p$ is at distance $W/4$ from either boundary of the block $B_j$ which contains it if that block is not the first or last block, and if $B_j$ is the first or last block then $i$ is well balanced if it is distance at least $W/4$ from the only block adjacent to $B_j$. It is easy to see that, in either case, a repetition $i$ is balanced for $p$ with probability at least $1/2$, taken only over the random choice of $r \in \{W/2, \ldots, W\}$. By Chernoff bounds, it follows that at least $R/3$ repetitions will be balanced for $i$ with probability at least $1 - 1/n^3$, which we condition on now. In what follows, we restrict our attention only to the balanced repetitions, and show that $\left| \mathcal{N}_{\hat{G}_p}^2(p) \right| \geq (1 - \delta)k$ holds even restricted to just the edges found in those repetitions.

By definition of the set $\mathcal{A}_p$ and a union bound over $n$ points and at most $R \leq n$ repetitions, for each (balanced) repetition $i$ there exists an $\ell = \ell_p \leq M$ such that simultaneously for all $x \notin \mathcal{A}_p$, we have $H_i^\ell(p) \neq H_i^\ell(x)$ with probability at least $1 - 1/n^2$. Call the event that this holds $E_p$, and condition on it for the remainder of the proof. Next, for each $x \in N_k(p)$, note that $H_i^\ell(p) = H_i^\ell(x)$ with probability at least $n^{-\rho}$. Moreover, the above analysis on the probability that a repetition $i$ only took randomness over the choice of the random $r \in \{W/2, \ldots, W\}$, it follows that the event that both $i$ is balanced and $H_i^\ell(p) = H_i^\ell(x)$ is at least $n^{-\rho}/2$. Thus, since we take $R = c_1 n^\rho \log n$ repetitions, taking $c_1 > 50$ with probability at least $1 - n^{-10}$ we will have $H_i^\ell(p) = H_i^\ell(x)$ for at least one balanced repetition $i$. Call this event $E_2$ and condition on it for the remainder of the proof. We now split into cases based on the two branches of the if-statement in Stars 2.

**Case 1:** $k > R/\delta$. For any balanced repetition $i$, we say that $i$ is bad if

$$\left| \{ x \in N_k(p) \mid H_i^{\ell_p}(p) = H_i^{\ell_p}(x) \} \right| \leq \frac{\delta k}{R} = \frac{\delta k}{c_1 n^\rho \log n}$$

We call a repetition good if it is not bad. Now for any point $x \in N_k(p)$, we say that $x$ is good if there exists a balanced repetition $i$ such that $H_i^{\ell_p}(p) = H_i^{\ell_p}(x)$ and $i$ is good. For such a good repetition $i$, we say that $x$ is good at $i$. In what follows, we fix any point $x \in N_k(p)$. We now prove the following claim.

**Claim B.4** *If $x \in N_k(p)$ is good at $i$, then with probability at least $1 - n^{-10}$, either $x$ is added to $\mathcal{N}_{\hat{G}_p}^2(p)$ on repetition $i$, or we add at least $k$ unique points from $\mathcal{A}_p$ to $\mathcal{N}_{\hat{G}_p}^2(p)$ on repetition $i$.*

**Proof:** To see this, for any point $y$, define $\sigma_y \in \{1, 2, \ldots, n\}$ be the ranking of $y$ in the lexicographical ordering induced by the set of hash functions $H^i$ corresponding to the $i$-th repetition. Let $t_- \le t_p$ be the smallest possible index such that the point $y$ with $\sigma_y = t_-$ satisfies $H_i^{\ell_p}(p) = H_i^{\ell_p}(y)$, and similarly define the point $t_+ \ge t_p$ as the largest such index. Note that, by conditioning on $E_p$, for every $y$ such that $\sigma_y \in [t_-, t_+]$, we have $y \in \mathcal{A}_p$.

First suppose that either $t_- < t_p - W/4$ or $t_+ > t_p + W/4$. Then since repetition $i$ is balanced, it follows that there are at least $W/4 > k$ points $y \in \mathcal{A}_p$ which are contained in the same block as $p$. Since we sample $s = c_2 \delta^{-1} n^\rho \log^2 n$ random leaders in this block, taking $c_2 > 100$, with probability larger than $1 - 1/n^{10}$ we will sample at least one $y \in \mathcal{A}_p$ which in that window. Then for any other $z \in \mathcal{A}_p$ which also fell in the same block, we create a path $(z, y), (y, p)$ of length two, each of which consists of only edges between points in $\mathcal{A}_p \cup \{p\}$. Thus, in this case we add at least $W/4 > k$ unique points from $\mathcal{A}_p$ to $\mathcal{N}_{\hat{G}_p}^2(p)$.

Otherwise, we have that $t_p - W/4 < t_- < t_p < t_+ < t_p + W/4$. Again, since repetition $i$ is balanced, it follows that every $y$ with $\sigma_y \in [t_-, t_+]$ falls in the same window as $p$. Since we know that repetition $i$ is good, it follows that $t_+ - t_- > \frac{\delta k}{c_1 n^\rho \log n}$. Thus, again by sampling $s = c_2 \delta^{-1} n^\rho \log^2 n$ leaders for $c_2$ larger than some constant times $c_1$, again with probability larger than $1 - 1/n^{10}$ we will sample at least one $y$ such that $\sigma_y \in [t_-, t_+]$ as a leader, implying that $y \in \mathcal{A}_p$. Conditioned on this, since $x$ must also fall in the same window, we add the edges $(x, y), (y, p)$ to $\hat{G}_p$, thereby adding $x$ is added to $\mathcal{N}_{\hat{G}_p}^2(p)$, which completes the proof of the claim. $\qquad\square$

By the above claim and a union bound, with probability at least $1 - n^{-9}$ it holds that for every $x \in N_k(p)$, if $x$ is good then either we add $x$ to $\mathcal{N}_{\hat{G}_p}^2(p)$ or we already have that $\left|\mathcal{N}_{\hat{G}_p}^2(p)\right| \ge k$. To complete the proof for Case 1, it suffices to show that the number of good points $x \in N_k(p)$ is at least $(1 - \delta) \cdot k$. To see this, note that if point $x$ is not good, then for every balanced repetition $i$ such that $H_i^{\ell_p}(p) = H_i^{\ell_p}(x)$ (for which there is at least one due to conditioning on $E_2$), we have $\left|\{x \in N_k(p) \mid H_i^{\ell_p}(p) = H_i^{\ell_p}(x)\}\right| \le \delta k / R$. For each bad point, we charge it's "badness" to the first balanced repetition $i$ for which $H_i^{\ell_p}(p) = H_i^{\ell_p}(x)$ and such that $i$ was bad. Since each bad repetition can be charged for at most $\delta k / R$ bad points, it follows that there can be at most $R \cdot \delta k / R < \delta k$ bad points in $N_k(p)$, and therefore the number of points good points $x \in N_k(p)$ is at least $(1 - \delta) \cdot k$, which completes the proof of Case 1.

**Case 2:** $k \le R/\delta$. In this case, we will demonstrate a stronger claim: for any $x \in N_k(p)$, and any balanced repetition $i$ such $H_i^{\ell_p}(p) = H_i^{\ell_p}(x)$ (for which there is at least one due to conditioning on $E_2$), either either the edge $(x, p)$ is added $\hat{G}_p$ on repetition $i$, or for at least $k$ unique points $y \in \mathcal{A}_p$ we add the edge $(y, p)$ points to $\hat{G}_p$ on repetition $i$. Note that, given this claim, the stronger fact that $\left|\mathcal{N}_{\hat{G}_p}^1(p)\right| \ge k$ in Case 2 follows immediately.

To see this, fix any $x \in N_k(p)$, and any balanced repetition $i$ such $H_i^{\ell_p}(p) = H_i^{\ell_p}(x)$. First suppose that $|\sigma_x - \sigma_p| \ge W/4$, WLOG we have $\sigma_x \ge \sigma_p + W/4$ (note that for the edge cases where $p$ is in the first or last block, there may be only one choice between $\sigma_x < \sigma_p$ or $\sigma_x > \sigma_p$ in this setting). Since $H_i^{\ell_p}(p) = H_i^{\ell_p}(x)$, it follows that every $y$ with $\sigma_x < \sigma_y < \sigma_x + W/4$ also satisfies $H_i^{\ell_p}(p) = H_i^{\ell_p}(y)$, and therefore $y \in \mathcal{A}_p$. Since $i$ is balanced, it follows that all such $y$ are in the same block as $p$, and therefore we created at least $W/4 > k$ such edges $(y, p)$ for points $y \in \mathcal{A}_p$. In the second case, assume $|\sigma_x - \sigma_p| \ge W/4$. In this case, by the balancedness of repetition $i$, it follows that $x, p$ are in the same block, and therefore we connect $(x, p)$ on this repetition, which completes the proof for case 2.

**Bounding The Number of Edges.** In case 1, for each of the $R$ repetitions, we create at most $2ns$ edges, thus the total number of edges is $O(Rns) = \tilde{O}(\delta^{-1} n^{1+2\rho})$. In Case 2, each vertex has degree at most $W$, so we create at most $Wn = O(kn) = O(nR/\delta)$ edges, thus the total number of edges is $RWn = \tilde{O}(\delta^{-1} n^{1+2\rho})$ as desired.

**Bounding The Runtime.** Note that evaluating the $M$ hash functions and sorting points lexicographically based on those hash functions requires $O(nM \log n + nM \cdot Time(\mathcal{H}))$ time. Thus, the overall runtime to construct all blocks used by the algorithm over $R$ repetitions is $\tilde{O}(RnM \cdot Time(\mathcal{H})) = \tilde{O}(RnM \cdot Time(\mathcal{H}))$. Once the blocks are complete, the remaining runtime is within a constant of $|E(G)|$, which is $\tilde{O}(\delta^{-1} n^{1+2\rho})$ as above, which completes the proof. $\square$

## C  Additional Implementation Details

### C.1  SortingLSH at Scale

The SortingLSH algorithm involves computing $R$ sketches per point, then sorting the $nR$ total sketches before subdividing into contiguous buckets with window size $W$. To achieve this at billion-point scales, we leverage the TeraSort algorithm [36].

### C.2  Training a Pairwise Similarity Model

Following [25], we train a similarity model on the task of distinguishing pairs of nodes which are in the same category from pairs of nodes which are in different categories. The model architecture uses shared-weight embedding towers to learn a symmetric representation of the node-level features, which are converted into a pairwise embedding using the Hadamard product. This embedding is concatenated with additional pairwise features, then used as input for a feed-forward neural net that generates the final binary prediction. The unthresholded scalar output of this final network provides a similarity score for any pair of nodes.

## D  Additional Experiments

### D.1  Additional Information of Datasets

- Each data item of MNIST is a $28 \times 28$ image of a hand written digit. There are total $10$ different digits and thus MNIST has $10$ ground truth clusters.

- Each data item of Wikipedia is an article. The feature of a data item is the set of words appeared in the article together with the frequency of appearance of each word.

- Amazon2m consists of products available on the Amazon web site. Each product has a $100$-dimensional real valued feature vector and co-purchase relationships with other products. The products are in $47$ categories and thus it has $47$ ground truth clusters.

- Random1B and Random10B are generated from mixture of Gaussians. Each data point has $100$ dimensions. The number of modes is $100$ where the $i$-th mode is a Gaussian distribution with mean $(0, 0, \ldots, 0, 1, 0, \cdots, 0)$ (the $i$-th entry is 1), and standard deviation $0.1$ of each entry. Each data point is drawn uniformly at random from a mode.

### D.2  Detailed Sketching Parameters

We set number of sketches $R = 25, 100, 400$ for all LSH and SortingLSH based algorithms. For Stars, if not specified otherwise, we set number of leaders $s = 25$.

For SortingLSH-based algorithms: we set window size $W = 250$, sketching dimension $M = 30$, and the maximum allowed bucket size to be $20000$ for all experiments.

For LSH-based algorithms:

- For non-Stars version of LSH based algorithm, we set the maximum allowed bucket size to be $1000$. For Stars version of LSH based algorithm, we set the maximum allowed bucket size to be $10000$.

- We employ SimHash of sketching dimension $M = 12$ for MNIST and $M = 16$ for Random1B and Random10B.

- For the Wikipedia dataset, we use the weighted Minhash with sketching dimension $M = 3$.

- For the Amazon2m dataset, we use a mixture of SimHash and MinHash with sketching dimension $M = 12$, i.e., randomly select each bit of hash value generated from SimHash or MinHash (it is easy to verify that the mixture of SimHash and MinHash satisfies Definition 2.1 for the similarity which is a mixture of cosine similarity and Jaccard similarity).

## D.3  Similarity for Amazon2m Learned by Neural Network

The embedding tower for each node combines the product embedding and a 1-hot encoded vector of co-purchased products, hashed to 1000 buckets. The Hadamard product of these embeddings is concatenated with: the cosine similarity of the product embeddings, an indicator variable conditioned on two products being copurchased, and the Jaccard similarity of the two products' copurchase sets. The embedding towers use two hidden layers of size 100 with ReLU activations [35]. The final prediction is made using the concatenated pairwise features and embedding Hadamard product with another MLP having two hidden layers of size 100 and ReLU activations again. The model is trained on all pairs which fall into an LSH bucket from a sample of 500,000 nodes from the original dataset and achieved an AUC of 0.92 on a holdout set from a distinct sample of nodes for the same-category task.

## D.4  Experiments of Different Number of Leaders

In this section, we study the impact of number of leaders. In particular, we fix the number of sketches $R = 400$ and evaluate the number of leaders $s = 1, 5, 10, 25$.

**Number of comparisons.**  In Figure 5, we show the number of pairwise similarity comparisons of each algorithm on MNIST, Wikipedia and Amazon2m datasets. As we can observe, Stars yields $\sim 100$-fold improvement in number of comparisons over the other algorithms when the number of leaders $s$ is chosen to be $1$ or $5$.

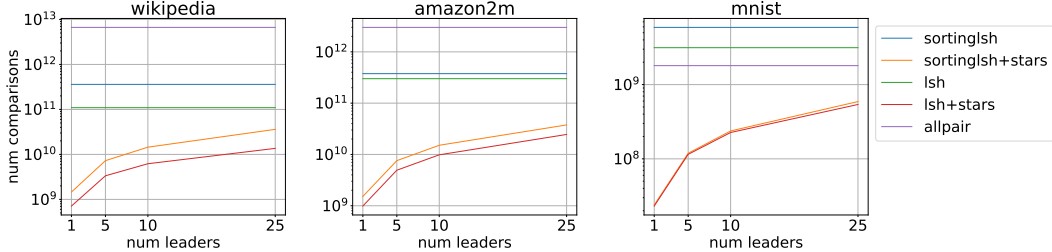

Figure 5: Number of comparisons of each algorithm on Wikipedia, Amazon2m and MNIST datasets. If we choose the number of leaders $s$ to be less than $25$, we can get better than $\sim 10$-fold improvement in the number of comparisons. In particular, if we choose $s = 1$ or $s = 5$, we have $\sim 100$-fold improvement in the number of comparisons.

**Coverage of Near(est) Neighbors.**  We evaluate the number of (approximate) near(est) neighbors which can be found for each point in Wikipedia, Amazon2m and MNIST, and by each algorithm with different number of leaders. For all algorithms, the number of sketches $R = 400$. The evaluation metric is the same as Section 5.

In Figure 6, we report the average of each ratio over all data points. When we choose more leaders, we obtain better recall.

We obtain sparser graphs when we use smaller number of leaders. Again, note that the graphs built by SortingLSH-based algorithms have the same sparsity since we only keep the $250$ closest points for each node (even for SortingLSH+Stars). The sparsity of graphs built by LSH-based algorithms is presented in Figure 7.

## D.5  Running Time

In this section, we investigate the running time of several experiments. We use *total running time* to refer to the summation of running time of building edges over all machines. We use *real running time*

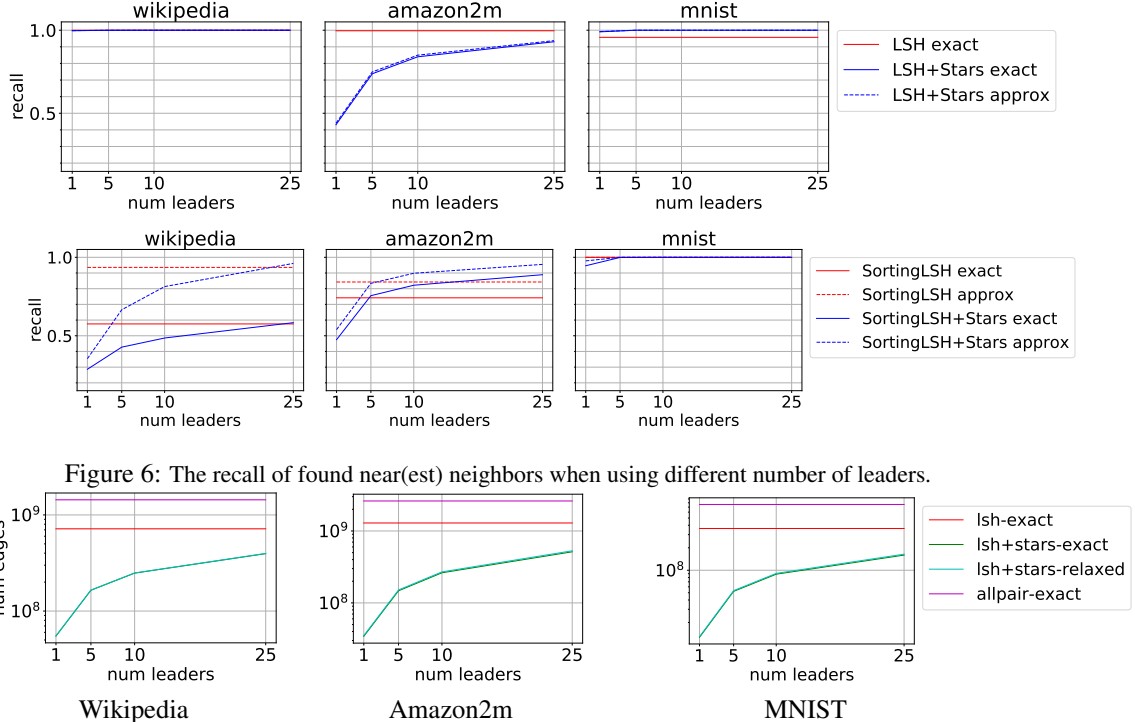

Figure 6: The recall of found near(est) neighbors when using different number of leaders.

Wikipedia               Amazon2m               MNIST

Figure 7: The number of edges with similarity at least 0.5 (0.495 for relaxed threshold) built by each algorithm with different number of leaders.

to refer to the actual running time of the entire job, which includes I/O, shuffle and scheduling time etc. Since real running time depends on the number of machines assigned to the job, I/O latency, and network condition etc., the total running time is usually less noisy. For all experiments, the worker pool contains 1000 number of machines (i.e., the maximum parallelism uses 1000 machines, though a job might not get all of them due to scheduling).

**Effect of similarity function.** We report the relative total running time of Stars versions of algorithms and non-Stars versions of algorithms for Amazon2m for both mixture of similarities and the similarity learned by neural networks. See Table 1 for LSH-based algorithms, and see Table 2 for SortingLSH-based algorithms.

**Experiments on 1B and 10B datasets.** For Random1B and Random10B datasets, we only run algorithms with number of sketches $R = 25$. We report the relative total running time of Stars versions of algorithms and non-Stars versions of algorithms for both random1B and random10B datasets in Table 3.

We also saw the speed-up of real running time on Random1B and Random10B datasets:

Table 1: Relative total running time of LSH-based algorithms for Amazon2m dataset for different similarity functions. The actual total running time corresponding to relative total running time 1.00 is $\sim 20$ hours. Note that the total running time is the summation of running time of computing edges over all machines.

|  | Mixture of similarities | Learned similarity |
|---|---|---|
| LSH+non-Stars (# sketches $R = 25$) | 1.00 | 3.34 |
| LSH+non-Stars (# sketches $R = 400$) | 11.05 | 49.75 |
| LSH+Stars (# sketches $R = 25$) | 0.05 | 0.51 |
| LSH+Stars (# sketches $R = 400$) | 0.87 | 4.39 |

Table 2: Relative total running time of SortingLSH-based algorithms for Amazon2m dataset for different similarity functions. The actual total running time corresponding to relative total running time $1.00$ is $\sim 38$ hours. Note that the total running time is the summation of running time of computing edges over all machines.

|  | Mixture of similarities | Learned similarity |
|---|---|---|
| SortingLSH+non-Stars (# sketches $R = 25$) | 1.00 | 22.39 |
| SortingLSH+non-Stars (# sketches $R = 400$) | 12.11 | 126.16 |
| SortingLSH+Stars (# sketches $R = 25$) | 0.23 | 2.65 |
| SortingLSH+Stars (# sketches $R = 400$) | 1.49 | 16.38 |

Table 3: Relative total running time of algorithms on Random1B and Random10B datasets. The actual total running time corresponding to relative total running time $1.00$ is $\sim 2632$ hours. Note that the total running time is the summation of running time of computing edges over all machines.

|  | Random1B | Random10B |
|---|---|---|
| LSH+non-Stars (# sketches $R = 25$) | 1.000 | 13.885 |
| SortingLSH+non-Stars (# sketches $R = 400$) | 0.076 | 0.907 |
| LSH+Stars (# sketches $R = 25$) | 0.017 | 0.178 |
| SortingLSH+Stars (# sketches $R = 400$) | 0.011 | 0.118 |

1. For Random1B: All jobs of Stars-versions of algorithms finished in 1 hour. SortingLSH+non-Stars finished in 1.5 hours and LSH+non-Stars finished in 2 hours.

2. For Random10B: All jobs of Stars-versions of algorithms finished in 2 hours. SortingLSH+non-Stars finished in 2 hours and LSH+non-Stars finished in 23 hours.

For large datasets, the running time of finding edges dominate other overheads and thus we can observe a large speed-up as described above.