# OpenReview forum: "Stars: Tera-Scale Graph Building for Clustering and Learning"
_NeurIPS.cc/2022/Conference — NeurIPS 2022 Accept_

### Official Review · Reviewer_kY9Y · 2022-06-27

**Rating:** 7
**Confidence:** 4
**Soundness:** 3 good
**Presentation:** 3 good
**Contribution:** 3 good

**Summary:**

Given a set of points X endowed with a measure of pairwise similarity, a similarity graph G(X,E) on X is supposed, loosely speaking, to draw edges between similar points and avoid edges between dissimilar points. Common variants define that a pair of points x,y should be neighbors if their similarity is above a fixed threshold, or if y is among the k most similar points to y for some fixed k. These graphs are commonly used in many applications. The computational challenges they pose are (a) that constructing them can be very costly, especially for very large datasets where exact construction is infeasible, and (b) that they need to be reasonably sparse since this affects their usability in the downstreams applications they are constructed for.

This paper advocates constructing these graphs such that similar points are not necessarily connected by an edge, but with a path of length 2. This addresses challenge (b) since this relaxed requirement allows the graph to be more sparse, and addresses challenge (a) since it lends itself to a very simple construction: bucketing the points in a way that roughly preserves similarity (there is a vast literature on efficiently generating such bucketing for large high-dimensional point sets, notably, by LSH), and then choosing a representative point from each bucket, and drawing an edge between the representative and the other point in the bucket (i.e., putting a star graph on the bucket). The algorithms in this paper repeat this with several random partitions to generate the similarity graph.

This approach to constructing similarity-preserving graphs has been suggested and studied before, but this paper applies it more specifically in the context of near-neighbor preserving graph construction, shows it can be implemented in practice on a very large scale, and includes an empirical evaluation. This adds a different angle to the existing literature (previous work that I am aware of has been strictly theoretical and focused on other flavors of similarity preserving graph).



**Questions:**

The paper is perhaps missing some discussion on what we lose when we allow similar points to be connected by a 2-hop path instead of an edge. What downstreams tasks is it suitable for, and are there downsteams tasks for which it is less suitable, or perhaps requires possibly costly modifications to those tasks (for example, replacing the greedy traversal used in graph-based NNS with 2-hop lookahead traversal). These questions come up naturally from the approach suggested in the paper, though I am generally content with largely leaving them for future work.

Another question is about guarantees for dissimilar points. The constructions in the paper ensure that each point is likely to have sufficiently many similar points in its 2-hop neighborhood, and the first construction also ensures that adjacent points (i.e., 1-hop away) are at least somewhat similar, but in neither construction I am not sure if there is a guarantee that dissimilar points are mostly disincluded in the 2-hop neighborhood. There is a sparsity guarantee on the total number of edges, which means that things are okay on average, but there could still be particular points where the 2-hop neighborhood contains many irrelevant point along with the near neighbors, which could cause latency issues (long runtime for specific queries) if these nodes participate in the downstream query (for example if they are visited in a traversal), if I am understanding this correctly.


**Strengths And Weaknesses:**

This is a good quality paper that seems to me like a solid contribution to the literature on large-scale NNS graph construction, is written well and clearly (up to some technical points noted below), and to me clears the bar for acceptance. The analytic novelty is perhaps not on the higher side, as the analysis directly utilizes classical and existing LSH techniques. However, the application to similarity graphs -- especially in practice and in demonstrably large scale -- is interesting and significant, and well-executed in this paper. This may advance the way we use LSH in large-scale settings and impact the way we construct large-scale similarity graphs. There is perhaps some room for more discussion and exploration of the usability and limitations of this approach, see "questions" below.

The technical writing could be more careful, with some undefined (on not clearly defined) notation and some lax writing in parts of the proofs (though there is no concern about correctness). These glitches can be corrected from context or if one is already familiar with the background literature, but may still impede readability for newcomers, which is a bit unfortunate since the paper is otherwise very well written. For example:
- tau_k(p) not explicitly defined (is line 199 supposed to be implicitly defining it as the similarity to the kth most similar point?)
- Incompatible neighborhood notation between lines 149 and 261
- The proof of proposition 3.3 makes use of properties of Simhash and Minhash that are never stated or referenced (the connection between the collision probability to the similarity)

and some more. I cannot list all of these but making the technical parts clearer and self-contained would likely help make the paper accessible to a wider audience.

Small corrections:
1. missing subscripts on the pi's in line 220
2. Did you mean d(p,q) instead of mu(p,q) on line 251

---

> ### Author Response · Authors · 2022-08-02
> **Author Response to Reviewer kY9Y**
>
> We thank the reviewer for their detailed appraisal and comments.
>
> With regards to the question of which downstream tasks two-hop spanners are suitable for, for any task which involves the propagation of information beyond two-hops in the graph (e.g. clustering tasks and label propagation), two-hop spanners would make a fine choice. At a high-level, running “two-rounds” of message propagation across edges in a two-hop spanner should be roughly equivalent to running a single round in a standard one-hop similarity graph. Since many downstream tasks do indeed involve aggregating neighborhood information beyond a single hop, two-hop spanners are widely applicable as a tool for these tasks. For the clustering and k-NN retrieval problems we evaluate empirically in the paper, we demonstrate that quality losses from moving to a two-hop spanner are negligible, demonstrating that these problems represent good opportunities to use two-hop spanners.
>
> With regards to the second question about guarantees for dissimilar points – this is an important question, and of course it should be a property of any good graph sparsifier, otherwise the definition of the spanner would not be useful. Indeed, this is true for two-hop spanners. Specifically, we obtain converse guarantees for neighbors in 2-hop distances just by use of the triangle inequality (using the distance version of the relevant similarity measure). For instance, if mu(x,y) is the Jaccard Similarity, then 1-mu(x,y) is the Jaccard distance. The first condition of Definition 2.4 says that every edge (x,z) has distance at most 1-r_1, so if there is a two-hop path x->z->y, it implies that d(x,y) < 2*(1-r_1), or that their similarity is at least 1-2*r_1. A similar guarantee holds for the second definition (using Stars + SortingLSH), except the thresholds that we use differ for each point (e.g., in post-processing we can remove all but the top-k closest neighboring edges of each point). Notice that every edge we create is weighted by the weight of that distance, so in the second definition it is possible to have two-far points (x,y) in a two-hop neighborhood x->z->y, but in this case either (x,z) or (z,y) will be far apart, and that will be accounted for in downstream tasks (such as HAC clustering) which take the weights of edges into account.

---

### Official Review · Reviewer_MbNY · 2022-07-09

**Rating:** 5
**Confidence:** 3
**Soundness:** 2 fair
**Presentation:** 2 fair
**Contribution:** 3 good

**Summary:**

This paper studies the similarity graph construction problem which is useful for downstream tasks such as clustering and graph learning. It proposes a scheme named Stars to construct two-hop spanners as a relaxation of the similarity graph by selecting leader points and generate star-shape edges. It claims it is superior because it requires at most $O(n^{1+O(\epsilon)})$ complexity in construction and the output two-hop spanners are sparser than normal similarity graphs.

**Questions:**

Q1. In two-hop spanners, similar points are two-hop neighbors, is that true reversely? That in Definition 2.4, for every $p \in N_2(q)$, is it guaranteed $\mu(p,q) \lq r_2$? If not, what is the impact on downstream tasks of introducing additional neighbors that are not similar?

Q2. What are the requirements (theoretical assumptions) for the LSH family $H$ in Theorem 3.4, especially for the sorting LSH technique? When migrating from the (r1 , r2 , rho)-sensitive families to (r1 , r2 , p1 , p2 )-sensitive families, are there any differences in theoretical and empirical performance?

Q3. Line 67 says Stars "construct significantly fewer than the O(n^1.99)-edges suggested by our theoretical guarantees." How is this presented in the experiments? Additionally, is this statement true for all scales of graphs, for example, small-scale graphs?

Q4. In Sec 5 "Sketching parameters" and "Clustering" tasks, how to determine which similarity / LSH is used for each dataset? Is there a general criterion for choosing similarity / LSH?

Q5. As stated in W4.3.

**Limitations:**

See W1 to W4.

**Strengths And Weaknesses:**

Strengths:

S1. This paper investigates a novel problem for graph construction on two-hop spanner graphs. It claims this type of graph has fewer edges and is hence more favorable for downstream tasks on large-scale data. It is of certain interest for the research in building sparse graphs from similarity representation, especially for large-scale data and distributed computation.

S2. This paper proposes a simple yet effective scheme Stars which can efficiently generate two-hop spanners based on similarity measures, i.e. Locality Sensitive Hash (LSH) Families. It provides theoretical analysis to show that Stars can construct approximate near neighbor (ANN) graphs under given error guarantees, and gives the time complexity bound.

Weaknesses:

W1. Given that two-hop spanner is a generalized type of similarity graphs, it is natural that algorithms designed on this graph type have better complexity than baselines. So whether the performance improvement comes from the Stars algorithm itself or is the merit of the graph relaxation is unclear. It is also not presented in the paper whether the difference in similarity graphs will result in performance drop in downstream tasks such as clustering (see Q1).

W2. As $\epsilon$ ranges from 0 to 1, I doubt whether the $O(n^{1+O(\epsilon)})$ complexity can be described as "nearly-linear". In the paper, it claims that the empirical time is significantly better than the theoretical bound, but this is not well explained (see Q3).

W3. For both theoretical and experimental evaluations, there are insufficient baselines included in this paper.

W3.1. For theoretical complexity analysis, it only presents the complexity of Stars without comparison to other approaches.

W3.2. For the graph construction experiment, it only compares brute force (all-pairs) and a non-Stars algorithm. What exactly is the "non-Stars algorithm" is unknown. Similar works like [21] or [R1] are also not evaluated.

W3.3. For the clustering experiment, there should be even more available baselines. I think it is even necessary to compare with non-LSH-based and non-similarity-based approaches.

W4. The experiment designs are still too simple beside the baseline issue. Sec 5 Empirical Study lacks critical information and is poorly written.

W4.1. It mainly evaluates the algorithm by changing the parameter $R$ of number of sketches. However, only having three points R = 25, 100, 400 is not enough. It is hard to observe the relation between complexity (num comparisons, num edges) and sketches.

W4.2. As complexity improvement is an important contribution in the paper, there should be experiments and analyses explicitly studying that. The evaluation of complexity on different scales of data is expected, rather than solely on two points of 1B and 10B data.

W4.3. Section 5 lacks necessary analysis and conclusion for experiments. For example in Fig 2, what is the meaning of the "recall" metric? Why the performance of "LSH exact" is different from others on MNIST? In Fig 4 Amazon2m, why "lsh+stars-learn" and "sortinglsh+stars-learn" seem to have lower performance than non-Stars methods?

Minor issues:
I1. Line 125 sentitive -> sensitive.

I2. Line 297-305, whether Random1B/Random10B dataset names are capitalized needs to be aligned.

I3. The log y axes in some figures (e.g. Fig 1 wikipedia, Fig 3 MNIST) are too rough to assess the results. Some data points (Fig 1 allpair, Fig 2 LSH+Stars exact, Fig 3 lsh+stars-relaxed) are hard to observe.

[R1] Zhang, Yan-Ming, et al. "Fast kNN graph construction with locality sensitive hashing." Joint European Conference on Machine Learning and Knowledge Discovery in Databases. Springer, Berlin, Heidelberg, 2013.

---

> ### Author Response · Authors · 2022-08-02
> **Author Response to Reviewer MbNY**
>
> We thank the reviewer for their detailed comments and suggestions. We respond to the main questions as numbered by the reviewer.
>
> W1: It is indeed true that the two-hop spanner relaxation allows for significantly fewer edges than standard (one-hop) similarity graphs – this is the key starting point of the paper. The main contribution of our paper is precisely to show that this relaxation does *not* result in quality or performances losses for downstream tasks. This is shown extensively in the experimental section, where we demonstrate that despite this significant reduction in edge/number of comparisons made, we still achieve comparable performance for k-NN and clustering recall.
>
> W2: Note that for c>/log(n)/loglog(n), a running time of n^{1+1/c} is indeed nearly linear. We generally think of c in the large-constant to mildly super-constant regime, in which case n^{1+1/c} = n^{1+o(1)} is known as “almost-linear” in the literature. We will specify this distinction, and when the algorithm is almost linear. Note that this runtime matches the (provably) best runtime for c-approximate nearest neighbor-search for the metrics discussed in the paper.
>
> W3.1: As mentioned in our response to Reviewer Gvmm, the related works on graph building do not build two-hop spanners, and therefore a direct theoretical comparison does not make sense – indeed, for the dataset example in our response to reviewer DHNc, any such (one-hop) algorithm would run in n^2 time since all edges above a threshold must be constructed. To the best of our knowledge, we are the first to theoretically analyze LSH-based two-hop spanners for graph building.
>
> W3.2: The non-Stars baselines are the standard LSH algorithm and SortingLSH (described in the "Sorting LSH" subsection of section 3.2.
>
> W3.3: In the clustering experiments, the purpose is to show that clustering quality does not degrade despite our graph containing only a subset of the edges. The clustering quality on the exact kNN graph is a hard upper bound in this sense, so additional comparisons would not add to the paper.
>
> W4.1 The relationship between num_comparisons and num_sketches is exactly linear (each sketch can be considered an independent parallel run of the algorithm with different random seed) so adding more points would not change the shape (note Fig 1 is a log plot).  num_edges increases monotonically with the number of sketches.
>
> W4.2 The complexity is linear in the size of the data, we demonstrate runtime on a total of 5 datasets (not just the two synthetic datasets).  For the final paper we can add a few more points for different synthetic dataset sizes to Fig 1 and 2.
>
> W4.3 See Q5 response
>
> Q1: Yes, this is true for any similarity measure which relates to a metric (e.g. Jaccard, cosine & Euclidean), see response to reviewer kY9Y. Thus, there is no loss (other than a factor of 2 from the triangle inequality) for downstream tasks.
>
> Q2: Nothing is lost in the translation between (r1,r2,rho)-families and ((r1,r2,p1,p2)-family – we choose to re-parameterize the family due to space constraints, because the reduction from the second to first is very standard in the literature. In theory and practice, one always first transforms a (r1,r2,p1,p2)-family into a (r1 , r2 , rho)-family before running nearest neighbor search, simply by concatenating multiple copies of the first hash function together, and then we have rho = log(1/p_1)/log(1/p_2). We will add additional details to the paper describing this reduction.
>
> Q3: In Figure 2, we show a good recall of finding 1.01-approximate near(est) neighbors in the 2-hop spanner constructed by our Stars algorithms. In the meanwhile, the number of edges shown in Figure 3 is much less than n^{1.99}. This phenomenon is observed in all 3 datasets used in Figure 2,3. MNIST is the smallest dataset which contains 60K data points. For dataset is smaller than MNIST, we can always say the number of edges is at most C*n^{1.99} for some large enough constant C. But we will clarify in the final version of the paper that our statement mainly serves for large datasets.
>
> Q4: the LSH family that is used is completely decided by the similarity that is used. For a specific similarity, one just uses the best known LSH for that similarity. For most metrics/similarities, such as L_2, L_1 cosine similarity, Hamming, Jaccard, etc, it is known exactly what the provably optimal LSH is. For the question of how to choose the metric, this is usually clear from context – if your data-points are normalized embeddings from a trained model, then one would use cosine similarity, and if they were (multi)-sets of objects (meta-data tags, bag-of-words), one would use Jaccard distance, ect.

---

> > ### Author Response · Authors · 2022-08-02
> > **Author Response to Reviewer MbNY (cont)**
> >
> > Q5: The detailed description of the “recall” metric is in the paragraph of “Coverage of Near(est) Neighbors” in Section 5. In particular, it is the ratio of (approximate) near neighbor points that can be found in 1 hop (or 2 hops depending on algorithms) in the graph. For MNIST, data points from the same class are close and thus a raw LSH bucket can be large. To prevent a huge number of pairwise comparisons in non-Stars, the buckets are randomly sub-partitioned (see last two paragraphs of Section 4). Thus, it misses a lot of near neighbor edges. However, this issue does not happen in the 2-hop spanner constructed by our Stars, which shows a better performance for MNIST dataset. For clustering on Amazon2m dataset (or any other datasets), since non-Stars computes the approximate kNN/similarity graph and Stars only outputs a relaxation graph, it is reasonable that the accuracy/quality of running the clustering via non-Stars is slightly better than via Stars. But since Stars runs much faster, as we mentioned in the paragraph “Effect of the similarity function” in Section 5: “given the same computational resources, switching to a Stars-based graph building strategy enables us to employ a wider range of similarity functions, and affords us the potential to significantly improve the quality of downstream tasks”. In particular, the message we want to convey is that (1) when using the same computational resources, we can run lsh+stars-learn instead of lsh-mix and get much better quality, (2) even if we afford more computational resources to run lsh-learn, the accuracy/quality is almost the same as lsh+stars-learn and has almost no improvement.

---

> > > ### Comment · Reviewer_MbNY · 2022-08-06
> > > **Response to author feedback**
> > >
> > > I would like to thank the authors for their detailed feedback. The authors stressed their contributions and addressed most questions, though the paper presentation is still expected to be improved in future versions. The last sentence in W3.2 is omitted in the response. Given the author feedback, I would like to raise my rating to "5: borderline accept".

---

### Official Review · Reviewer_Gvmm · 2022-07-12

**Rating:** 6
**Confidence:** 3
**Soundness:** 4 excellent
**Presentation:** 2 fair
**Contribution:** 3 good

**Summary:**

This work proposes an algorithm for efficiently constructing similarity graphs for extremely large graphs. This work introduces the idea of using two-hop spanners (derived from concept of metric spanners), in which all points with similarity greater than some threshold are guaranteed to be less than or equal to 2 hops apart, and no edge with some small similarity level have an edge between them. With the relaxed goal of similar points being within two hops as opposed to one, the algorithm is able to make significant gains in overall efficiency. They give an algorithm for constructing the two-hop spanner threshold graph using Locality Sensitive Hashing (LSH) as well as an algorithm using Sorted LSH for approximating K-nearest neighbors. They give algorithms and theoretical guarantees that their algorithm approximates nearest neighbors using angle or jaccard similarity measures, as well as theoretical guarantees on efficiency.

They confirm the theoretical results with experiments on a combination of real and synthetic datasets. The experiments include results on the number of comparisons made for various algorithms, how close the number of sketches hyperparameter affects approximation to KNN in terms of recall, as well as the coverage of the approximation. There is also an experiment showing the performance on clustering of various versions of the algorithm and the all-pairs baseline.

**Questions:**

1) Can the related work be improved to show how does this method compares to the most similar other LSH-based/approximate KNN and similarity graph construction algorithms?

2) Have you made any comparisons of theoretical guarantees and/or experimental results against other fast clustering/nearest neighbor algorithms?

**Limitations:**

Adequately done.

**Strengths And Weaknesses:**

Originality:

++ Somewhat original. Builds off techniques from previous similarity clustering algorithms as well as using techniques of locality sensitive hashing and two-hop spanners.

-- Not clear to me how this compares to other approximate K nearest neighbor and similarity graph construction algorithms

e.g.

Eiras-Franco, Carlos et al. “Fast Distributed kNN Graph Construction Using Auto-tuned Locality-sensitive Hashing.” ACM Transactions on Intelligent Systems and Technology (TIST) 11 (2020): 1 - 18.

Zhang, Yanming et al. “Fast kNN Graph Construction with Locality Sensitive Hashing.” ECML/PKDD (2013).

Wang, G. et al. “Learning to Prune: General and Efficient Approximate Nearest Neighbor Search with Direction Navigating Graph.” 2022 the 5th International Conference on Data Storage and Data Engineering (2022): n. pag.

Quality:

++ This is a complete, high quality work.

++ While I have not confirmed all the proofs in the appendix, the theorems seem to be soundly derived.

++ The experiments show the effectiveness of the algorithm in terms of efficiency and approximation.

-- It would be nice to have some more experiments showing the usefulness on downstream tasks.

-- Also, no comparisons to similar works.

Clarity:

++ While math heavy, the paper remains clear. As a nice benefit, the paper would still read pretty clearly even for a light read as many readers might be doing.

Significance:

++ It seems like this would be of interest to some readers. It seems certainly on the application side this would be of interest.

-- It would be nice to have more examples of downstream use cases. The clustering experiment is helpful in that regard but not super clear-cut. It is


Spelling/Gramar:

line 172: twp-hop -> two-hop

---

> ### Author Response · Authors · 2022-08-02
> **Author Response to Reviewer Gvmm**
>
> We thank the reviewer for their comments and questions. We address each of the questions and concerns below.
>
> -“Not clear to me how this compares to other approximate K nearest neighbor and similarity graph construction algorithms”:
>
> In some sense, our algorithm is orthogonal to other existing approximate k-nearest neighbor graph construction algorithms. Existing approximate k-nearest neighbor graph construction algorithms try to connect each point with all k (approximate) nearest neighbors while our constructed graph does not aim to directly connect each point to all k (approximate) nearest neighbors. Our algorithm preserves (approximate) k nearest neighbor information in 2 hops and is used for other down-stream tasks such as clustering. As benefits from the relaxation, we show in theory, our running time and the number of comparisons is near linear and independent from k, and we also get fast running time and small number of comparisons in practice. In contrast, most existing LSH based k (approximate) nearest neighbor graph construction may have much larger running time and number of comparisons in the worst case in theory. For example, both references Eiras-Franco, Carlos et al. and Zhang, Yanming et al. mentioned by the reviewer tried to improve the construction of the sketching buckets. However, they still need to run all-pair comparisons in each bucket which needs quadratic time in the bucket size for each bucket (the worst case bucket size depends on k, i.e., the worst case bucket size can be much larger for larger k). The Wang, G. et al. work mentioned by the reviewer requires an approximate k-nearest neighbor graph as input and thus is not relevant for comparison.
>
> -“It would be nice to have some more experiments showing the usefulness on downstream tasks”:
>
> Since clustering is one of the most important tools in unsupervised learning, we believe our experiments already show usefulness on such downstream tasks. In addition, we show how to apply the STARS  algorithm to handle clustering with different similarities including the similarities learned by neural networks. Furthermore, Affinity clustering with a large enough number of rounds is equivalent to connected components, thus our experiment also implies its usefulness on connected components of the similarity graph. We also show its usefulness on single-linkage clustering and connected components with theoretical guarantees (see discussions in appendix).
>
> -“no comparisons to similar works”:
>
> To the best of our knowledge, we are the first to study 2-hop spanners for graph building empirically. We tried our best to compare our results with the algorithms without applying the idea of 2-hop spanner.
>
> -“Can the related work be improved to show how this method compares to the most similar other LSH-based/approximate KNN and similarity graph construction algorithms?”:
>
> Note that the goal of our algorithm is not to construct an approximate kNN graph or similarity graph. The main message that we want to convey is: for many important downstream tasks such as clustering, we would obtain similar performance by running it on the 2-hop spanner instead of the kNN/similarity graph. At the same time, the construction time and number of comparison needed to construct our 2-hop spanners are significantly smaller than what is needed to build a standard kNN or similarity graph. Notice that when we did the evaluation of the accuracy/quality of some downstream tasks, we compared our algorithm with the ground truth kNN/similarity graphs (obtained by bruteforce all-pair comparisons).
>
> -“Have you made any comparisons of theoretical guarantees and/or experimental results against other fast clustering/nearest neighbor algorithms?”:
>
> We did not compare our clustering algorithms with other clustering/nearest neighbor algorithms since it is not our focus. Our goal is to show that for the same graph based downstream algorithms (such as graph based clustering), its accuracy/quality by running on the kNN/similarity graph is similar to the accuracy/quality by running on the 2-hop spanner. Meanwhile, we can efficiently construct 2-hop spanners in both theory and practice.

---

### Official Review · Reviewer_DHNc · 2022-07-14

**Rating:** 5
**Confidence:** 2
**Soundness:** 2 fair
**Presentation:** 2 fair
**Contribution:** 3 good

**Summary:**

The paper proposes methods to construct 2-hop spanners to approximate similarity graph in extremely large data. The idea is to use  hashing functions to reduce the neighborhood searching load.

**Questions:**

What is the benefit of 2-hop spanners compared to a similarity graph of: the same threshold (r_1), and larger threshold (somewhere between r_1 and r_2)?

**Ethics Review Area:**

["I don’t know"]

**Strengths And Weaknesses:**

I think the paper can be a significant contribution to the problem of similarity graph construction via using 2-hop spanners, which increase the sparsity of the graphs.

I could not check the maths in detail to approve the analysis.

What the paper's missing is to validate the benefit of 2-hop spanners compared to a similarity graphs as the two-hop spanners contain a similarity graph with higher threshold.

The paper has typos in many places.

---

> ### Author Response · Authors · 2022-08-02
> **Author Response to Reviewer DHNc**
>
> We thank the reviewer for their comments and questions. We would like to reply to the main question of the reviewer regarding the difference between two-hop spanners and (one-hop) similarity graphs which use a higher threshold. The two notions of similarity graphs are fundamentally different, with the key advantage, as discussed and evaluated in the paper, is that two-hop spanners are significantly sparser than traditional (one-hop) similarity graphs, regardless of the threshold used.
>
> For instance, consider a cluster of n points which are pairwise distance almost exactly 1 from each other – such a dataset is possible in dimension O(log(n)). Then a one-hop similarity graph with threshold lower than 1 would create a dense clique, with a very large n^2 number of edges. A threshold higher than 1 would create no edges. On the other hand, using two-hop spanners, one would construct a star graph, centered at an arbitrary point, and achieve a perfect 2-hop spanner, and would result in these points correctly being clustered/classified together in a downstream task. Thus, two-hop spanners can decrease the sparsity of the graph by a quadratic factor, with little to no loss in quality for downstream tasks, in a way that cannot be replicated by any one-hop threshold-similarity graph.

---

### Meta-Review · Area_Chair_6BAv · 2022-08-26

**Recommendation:** Accept
**Confidence:** Certain

**Metareview:**

All reviews for this paper were positive, albeit with a varying level of enthusiasm. Reviewers found the problem (large scale similarity search) to be important, and the authors contribution to it (faster search using 2-hops search graph constructed via  LSH) significant. There were some concerns about the baselines in experimental evaluation, and some (relatively minor) presentation issues. Ultimately, the positives significantly outweighed the negatives.

**Award:**

No

---

### Decision · Program_Chairs · 2022-09-14

Accept